

# Growth of atmospheric clusters involving cluster-cluster collisions: comparison of different growth rate methods

J. Kontkanen[1], T. Olenius[2], K. Lehtipalo[1,3], H. Vehkamäki[1], M. Kulmala[1] and K. E. J. Lehtinen[4]

[1]Department of Physics, University of Helsinki, FI-00014 Helsinki, Finland
[2]Department of Environmental Science and Analytical Chemistry (ACES) and Bolin Centre for Climate Research, Stockholm University, SE-10691 Stockholm, Sweden
[3]Paul Scherrer Institute, 5232 Villigen PSI, Switzerland
[4]Finnish Meteorological Institute and Department of Applied Physics, University of Eastern Finland, FI-70211 Kuopio, Finland

*Correspondence to*: J. Kontkanen (jenni.kontkanen@helsinki.fi)

**Abstract.** We simulated the time evolution of atmospheric cluster concentrations in a one-component system where clusters grow not only by condensation of monomers, but where also cluster-cluster collisions significantly contribute to the growth of the clusters. Our aims were to investigate the consistency of the growth rates of sub-3 nm clusters determined with different methods, and the validity of the common approach to use them to estimate particle formation rates. We compared the growth rate corresponding to particle fluxes (FGR), the growth rate derived from the appearance times of clusters (AGR) and the growth rate calculated based on irreversible vapor condensation (CGR). We found that the relation between the different growth rates depends strongly on the external conditions and the properties of the model substance. The difference between the different growth rates was typically highest at the smallest, sub-2nm sizes. FGR was generally lower than AGR and CGR; at the smallest sizes the difference was often very large, while at sizes larger than 2 nm, the growth rates were closer to each other. AGR and CGR were in most cases close to each other at all sizes. The difference between the growth rates was generally lower in conditions where cluster concentrations were high, and evaporation and other losses thus less significant. Furthermore, our results show that the conventional method used to determine particle formation rates from growth rates may give estimates far from the true values. Thus, care must be taken not only in how the growth rate is determined, but also in how it is applied.

## 1 Introduction

Atmospheric new particle formation has been observed to occur frequently in various environments around the world (Kulmala et al., 2004). The process has been estimated to significantly contribute to the global concentrations of cloud condensation nuclei (CCN), and thus affect the Earth's climate (Spracklen et al. 2008; Merikanto et al., 2009). The primary quantity characterizing new particle formation events is the particle formation rate, which is defined, for any size, as the flux of particles growing past that size (Kulmala et al., 2004). For determining this flux, the particle growth rate (GR) is commonly used (Kulmala et al., 2012).



With respect to analyzing and quantifying new particle formation events, GR has had several different interpretations and uses. Theoretically, GR for a given particle is straightforward to define: it is the rate at which the particle diameter changes at a given moment in time. However, as this growth is caused by random collisions of vapor molecules, GR can vary a lot in time and from particle to particle. Especially, all particles of the same size and chemical composition do not grow at exactly the same rate, as is inherently assumed in e.g. the condensational growth term in the standard version of the continuous aerosol general dynamic equation (GDE; e.g. Seinfeld and Pandis, 2006). Still, a mean size-dependent value can be derived for GR, resulting in the well-known expressions for the free-molecular and continuum regimes of condensational growth, as well as various interpolations for the transition regime (see e.g. Seinfeld and Pandis, 2006). These expressions have been used and are convenient when trying to estimate vapor concentrations from observed GR or vice versa (e.g. Dal Maso et al, 2005; Nieminen et al., 2012). In this article, such a growth rate is called the growth rate based on irreversible vapor condensation, abbreviated as CGR. Another important use of GR lies in relating it to the dynamics of the evolving size distribution as the population of particles undergoes condensational growth. It is used in this context especially when estimating so-called survival rates, i.e. the fraction of particles that are not scavenged by background particles, instrument walls or other sinks while growing to a certain size. In this case, it is natural to define GR at a specific diameter by the flux $J$ of particles growing past the given diameter (e.g. Wang et al., 2013; Olenius et al., 2014). In this article, such a growth rate is called the flux-equivalent growth rate FGR.

To study the first steps of new particle formation, the growth rates of small, sub-3nm particles have been deduced from experimental data using various methods. The CGR method has been applied to specific measurement conditions by using the observed concentrations of precursor vapors in the calculation (Nieminen et al., 2012). The growth rates of charged particles have been derived from ion spectrometer data by following the time evolution of the concentration maximum (Hirsikko et al, 2005, Manninen et al., 2009, Yli-Juuti et al., 2011). Measuring sub-3 nm electrically neutral particles is challenging, and therefore their growth rates have been indirectly deduced from the time lag between the rise in sulfuric acid concentration and the increase in the concentration of 3 nm particles (Weber et al., 1997; Sihto et al., 2006). However, recent instrumental development has enabled the detection of neutral clusters with mobility diameters of down to ~1 nm by using instruments such as the DEG-SMPS (Jiang et al., 2011) and the PSM (Particle Size Magnifier; Vanhanen et al., 2011). Consequently, the growth rates in the sub-3 nm size range have been assessed, for example, by fitting the size distributions measured with a DEG-SMPS to the GDE (Kuang et al., 2012), or by determining the times at which the onset of new particle formation is detected with CPCs with different cut-off sizes (Riccobono et al., 2012). Recently, the growth rates of sub-3nm particles and molecular clusters have been determined from the appearance times of clusters in different size bins measured by scanning the PSM (Kulmala et al., 2013; Lehtipalo et al., 2014) or the appearance times of specific clusters detected by mass spectrometers (Lehtipalo et al. 2014; 2015). The appearance time has generally been defined as the time at which the concentration in the size bin, or the signal intensity of the cluster, reaches 50% of its total increase. In this article, the growth rate derived from the appearance times is referred to as AGR.



To investigate the validity of the appearance time method, Lehtipalo et al. (2014) applied the method to particle size distribution data simulated with an aerosol dynamics model. They found that the growth rates determined from the appearance times were close to the average condensational growth rates used as input in the simulation. On the other hand, Olenius et al. (2014) took a different approach to assess the AGR method by using a cluster kinetics model that does not inherently assume any growth

rates, but simulates the evolution of the cluster population via discrete collisions and evaporations of molecules. They compared the growth rates obtained with the appearance time method (AGR) to the growth rates corresponding to the molecular fluxes (FGR) and concluded that AGR was higher than FGR in the studied conditions. The difference was largest for the smallest clusters and was often strongly affected by the ambient conditions. Although Olenius et al. (2014) showed that AGR and FGR may not be equal, they concentrated on an ideal situation where the growth proceeds only by monomer

collisions and evaporations. In reality, there are situations where collisions between two clusters may contribute significantly to the growth (Lehtipalo et al., 2015) and they should, therefore, be taken into account when calculating the flux-equivalent growth rate. Furthermore, Olenius et al. (2014) used a resolution of a single molecule in their analysis, which is not possible when analyzing experimental particle size distributions.

As GR can been interpreted and determined from experimental data in many different ways, it is essential to compare the

results obtained with different methods. In this study, we compare the three above mentioned growth rate definitions, FGR, CGR and AGR, by applying them to modelled size distribution data. We use the same dynamic model as Olenius et al. (2014) to simulate the time evolution of cluster concentrations in a one-component system. As opposed to the simulations done by Olenius et al. (2014), in our model system a significant part of clusters' growth proceeds via collisions of small clusters in addition to monomer attachments. Because the growth rate of a single cluster is ambiguous in this case, we group the clusters

into size bins for which we calculate the growth rates. This also makes our analysis resemble the analysis of measured particle size distributions with a size resolution poorer than one molecule. We estimate AGR as in Lehtipalo et al. (2014), FGR analogously with Olenius et al. (2014), and CGR directly from the vapor monomer concentration (Nieminen et al., 2012).

Our aim is to answer the following questions: 1) How important are cluster-cluster collisions for the growth of the cluster population in different conditions in our model system? 2) How consistent with each other are the flux-equivalent growth rate,

the growth rate derived from the appearance times of the clusters, and the growth rate calculated based on irreversible vapor condensation? 3) How valid is the conventional method used to estimate particle formation rates from growth rates? We examine these questions in different conditions by varying the saturation vapor pressure of the vapor, the vapor source rate and the magnitude of an external sink reducing the vapor and clusters. The simulated conditions correspond to the typical conditions observed during new particle formation in, for example, a boreal forest. In most of the simulations the size-

dependent evaporation rate is set to decrease monotonically with increasing cluster size, corresponding to increasing cluster stability. However, we also test a different evaporation profile in order to study the effect of elevated concentrations of stable small clusters on the growth of the cluster population. Furthermore, we investigate how the size resolution, i.e. the width of the size bins, affects the results.




## 2 Methods

### 2.1 Determining the growth rates

For the growth rate analysis, the clusters were grouped into size bins so that each bin contains an equal number of cluster sizes, i.e. in linear volume space the bins are of equal width. The time evolution of the total cluster concentration $C_i$ in a certain size bin $i$ can be described with the following equation:

$$\frac{dc_i}{dt} = J_{i-1,i} - J_{i,i+1} - S_i \tag{1}$$

Where $J_{i-1,i}$ is the flux coming to size bin $i$ from the previous bin $i-1$, $J_{i,i+1}$ is the flux from bin $i$ to bin $i+1$ and $S_i = \sum S_j$ is the total external sink for size bin $i$, summed over all cluster sizes $j$ that belong to bin $i$.

Equation 1 can be obtained directly by integrating the continuous general dynamic equation (GDE; Friedlander, 1977) for aerosols, however, including only the growth and sink terms. If traditional continuous approach is used and clusters are assumed to grow synchronously by condensation, we can write

$$J_{i,i+1} = n \cdot \mathrm{GR}\big|_{\text{at the boundary between bins } i \text{ and } (i+1)} \tag{2}$$

and similarly for $J_{i-1,i}$ (Lehtinen, et al. 2007). Here $n$ is the number concentration distribution function $dC/dd_p$ and $\mathrm{GR} = dD_p/dt$ the diameter growth rate of the clusters.

In principle, it seems straightforward to combine Eqs (1) and (2) to obtain a method to determine growth rates from size distribution data. However, the possible contribution of larger clusters to growth, and the need to somehow approximate $n$ and GR (at bin boundaries) complicate the task.

### 2.1.1 Flux-equivalent growth rate (FGR)

Here we follow the Eulerian approach used by Olenius et al. (2014) and referred to as the flux-equivalent growth rate (FGR). The method is based on defining GR by Eq. (2) even if the original underlying assumptions of Eq. (2) would not be valid. Furthermore, $n$ is approximated with the value at the lower side of the bin boundary, resulting in

$$\mathrm{FGR}_{i,i+1} = \frac{J_{i,i+1}}{n_i} = \Delta D_{p,i} \frac{J_{i,i+1}}{c_i}, \tag{3}$$

where $\Delta D_{p,i}$ is the width of bin $i$ in diameter space. These assumptions and approximations are generally made when treating experimental, size bin classified distribution data by using Eqs (1) and (2).

To obtain the net flux $J_{i,i+1}$ between size bins $i$ and $i+1$, we first determine the fluxes between different cluster sizes with the following method: a) each collision between cluster sizes $p$ and $q$ grows $p$ by the addition of the molecules in $q$ if $p > q$, contributing to the flux between $p$ and $p+q$. For $q$, such a collision is treated as a sink. b) If the colliding clusters are of the same size ($p = q$), one of them is considered to grow and the other to be lost. In addition, to obtain the net flux between cluster sizes $p$ and $q$, monomer evaporations and cluster fissions are taken into account. After calculating the net fluxes between each pair of cluster sizes, the net flux $J_{i,i+1}$ can be obtained by summing up the individual collision–evaporation fluxes crossing the



boundary between the bins. Note that when using this method it must be ensured that the growth is dominated by collisions of small enough clusters, and the bins are wide enough, so that the growth from bin $i$ to the bins larger than $i+1$ does not occur.

**2.1.2 Growth rate from appearance times of clusters (AGR)**

One possible way to assess growth rates from the time evolution of a particle distribution is based on the times at which concentrations in different size bins reach their maximum (Lehtinen and Kulmala, 2003). This is convenient for cases like nucleation bursts where there is a growing mode of particles. However, for cases where the system approaches a time-independent steady-state, this method obviously does not work. Here we investigate a method to obtain GR from appearance times of clusters (AGR) in the size bins, by defining the appearance time $t_{app,i}$ for the size bin with the mean diameter $D_{p,i}$ as the time at which the concentration of the bin reaches 50% of the total increase in the concentration in that size bin (Lehtipalo et al., 2014). Then, AGR is obtained from numerical differentiation of the ($t_{app}$, $D_p$)-data:

$$AGR_{i,i+1} = \frac{D_{p,i+1} - D_{p,i}}{t_{app,i+1} - t_{app,i}}. \tag{4}$$

In recent experimental studies, AGR has been determined by applying a linear fit to the ($t_{app}$, $D_p$)-data over a range of several instrumental size classes. As the aim of this work is to examine the particle flux and growth rate as a function of size, AGR is here determined separately for each size bin as in Eq. (4).

As one of the main reasons to estimate GR from particle size distribution dynamics is to estimate particle flux (or formation rate) $J$, we also compare the different methods in terms of fluxes. Thus, similarly to Eq. (3), we define a 'cluster appearance time flux' as follows:

$$J_{app,i,i+1} = \frac{AGR_{i,i+1}}{\Delta D_{p,i}} C_i \tag{5}$$

In the analysis of experimental particle distribution data, Eq. (5) is generally used to calculate the particle flux towards larger sizes from measured or calculated growth rate (Kulmala et al., 2012). Thus, comparing the flux $J_{app}$ to the real flux $J$, which is directly obtained from the simulations, provides information on how the use of Eq. (5) affects the conclusions of the data analysis.

In addition to the change in the mass diameter of the clusters, we determine FGR and AGR also as the change in the number of molecules, $N_{mols}$. These are obtained by replacing the width of the size bin $\Delta D_{p,i}$ in diameter space in Eqs (3) and (4) with the width of the bin $\Delta N_{mols}$ in number of molecules space.

**2.1.3 Growth rate by assuming irreversible vapor condensation (CGR)**

The kinetic hard-spheres collision rate between a vapor molecule with diameter $D_{mon}$ and a cluster with diameter $D_p$ is given by

$$K_{kin} = \frac{\pi}{4}(D_p + D_{mon})^2(\bar{c}_p{}^2 + \bar{c}_{mon}{}^2)^{1/2}(C_{mon} - C_e), \tag{6}$$





where $\bar{c}$ is the thermal speed, $C_{mon}$ is the vapor concentration and $C_e$ is the vapor concentration corresponding to the equilibrium vapor pressure over the cluster. When analyzing experimental data, it is usually assumed that saturation vapor pressure $p_{sat} = 0$, which means that the vapor is assumed to condense irreversibly. In this case, the growth rate of a cluster as a change of mass diameter is obtained from (Nieminen et al., 2012):

$$\text{CGR} = \frac{\gamma}{2\rho}(1 + \frac{D_{mon}}{D_p})^2(\frac{8k_BT}{\pi})^{1/2}(\frac{1}{m_p} + \frac{1}{m_{mon}})^{1/2}m_{mon}C_{mon}. \qquad (7)$$

Here $\rho$ is the condensed phase density, and $m_p$ and $m_{mon}$ are the masses of the cluster and the vapor molecule. $\gamma$ is a correction factor that needs to be added if CGR is calculated in continuum regime (see Nieminen et al., 2012). When applying Eq. (7) for calculating CGR for different size bins in this study, we use the diameter and mass of the cluster at the upper limit of each size bin for $D_p$ and $m_p$.

In addition to the fact that CGR calculated from Eq. (7) takes into account only monomer collisions and no evaporation, the essential difference between CGR and FGR is the perspective from which the growth is studied. CGR corresponds to the traditional Lagrangian approach, where the growth of an individual cluster between different size bins is followed. FGR, on the other hand, corresponds to the Eulerian approach, where the net flux between adjacent size bins is studied. See Olenius et al. (2014) for further discussion about the differences between these approaches.

## 2.2 Simulations

Our model substance was assumed to consist of spherical molecules and clusters with the properties of sulfuric acid: a molecular mass ($m_{mon}$) of 98.08 amu, a liquid density ($\rho$) of 1830 kg m$^{-3}$ and a surface tension ($\sigma$) of 0.05 N m$^{-1}$. However, the saturation vapor pressure of the model substance was lowered from that of sulfuric acid to decrease the evaporation rate of clusters. This qualitatively mimics the stabilization of sulfuric acid clusters by base molecules, such as ammonia or amines (e.g. Kurtén et al., 2008). The simulations included clusters with 1 to 70 molecules. The clusters were grouped into size bins containing an equal number of clusters (in most simulations ten), for which fluxes and growth rates were determined. The Gibbs free energy of formation of the clusters was calculated from the classical one-component liquid droplet model to obtain a qualitatively realistic evaporation profile. In the majority of the simulations the Gibbs free energy profile had a single maximum and no minima as a function of cluster size. This corresponds to a monotonically increasing stability with increasing cluster size in the studied size range. In addition, a set of simulations was performed using a free energy profile with lowered formation free energies for the smallest clusters (see Fig. A1 in Appendix). This corresponds to a system with elevated concentrations of small stable clusters, similarly as in the simulation study by Vehkamäki et al. (2012) and possibly also in the atmosphere (Kulmala et al., 2013). Collision rates between clusters were obtained from Eq. (6) with $C_e = 0$, and cluster evaporation rates were calculated from the Gibbs free energies of formation of the clusters (e.g. Ortega et al., 2012). The external losses were assumed to depend on the cluster size according to (Lehtinen et al., 2007):

$$S(D_p) = S(D_{mon}) \times \left(\frac{D_p}{D_{mon}}\right)^b \qquad (8)$$




Here $D_{mon}$ is the diameter of a monomer. The exponent $b$ was set to 1.6, in which case Eq. (8) corresponds to the typical size-dependency of losses caused by background aerosol particles in a boreal forest (Lehtinen et al., 2007). In all the simulations the temperature was set to 278 K. The initial cluster concentrations were set to a steady-state distribution at a monomer concentration of $5 \times 10^5$ cm$^{-3}$ s$^{-1}$.

A summary of the performed simulations is presented in Table 1. In the first four simulation sets, the Gibbs free energy profile had one maximum and no minima. In the first simulation set, the effect of monomer concentration was studied: a constant source of monomer was assumed so that the final steady-state monomer concentration was $10^6$–$10^7$ cm$^{-3}$. These monomer concentrations are of the same order of magnitude as sulfuric acid concentrations observed during new particle formation in a boreal forest (Kulmala et al., 2013). The reference loss coefficient $S(D_{mon})$, describing the external sink, was set to $10^{-3}$ s$^{-1}$,

which is of the order of magnitude of the loss of clusters onto pre-existing particles in a boreal forest (Dal Maso et al., 2005) or walls in a chamber experiment. A saturation vapor pressure of $2 \times 10^{-10}$ Pa was used to study the situation where small clusters significantly contribute to the growth of the cluster population.

In the second simulation set, the aim was to investigate the effect of cluster stability on the growth of the cluster population by varying the saturation vapor pressure, to which the evaporation rates are directly proportional, from $1.5 \times 10^{-10}$ Pa to $1 \times 10^{-9}$

Pa. The final steady-state monomer concentration was set to $5 \times 10^6$ cm$^{-3}$.

In the third simulation set, the effect of the magnitude of the external sink was studied by setting the loss coefficient to 0.7 $\times 10^{-3}$–$2 \times 10^{-3}$ s$^{-1}$. The monomer source rate was set to $5.5 \times 10^3$ cm$^{-3}$ s$^{-1}$, which produces the steady-state monomer concentration of $5 \times 10^6$ cm$^{-3}$ when the loss coefficient is $10^{-3}$ s$^{-1}$ and the saturation vapor pressure is $2 \times 10^{-10}$ Pa.

In the fourth simulation set, we studied how the width of the size bins affects the growth rates by varying the size bin width

from 5 to 14 clusters. Furthermore, as we wanted to compare our results directly with the results of Olenius et al. (2014) who used an ideal precision of one molecule in their simulations, we performed additional simulations with a cluster population that grows only by monomer additions. In these simulations, we set the saturation vapor pressure of the model substance to $1 \times 10^{-9}$ Pa, and allowed only monomer collisions and evaporations in our system.

The fifth simulation set was otherwise identical with the first simulation set, but the Gibbs free energy profile was different: a

negative term of $90 \times (\exp(-(n_{mols} - 1)/4.2) - \exp(-(n_{mols} - 1)/4.5))$ was added to the classical expression for the free energy in order to decrease the formation free energies of the smallest clusters while keeping the free energies of larger clusters unchanged. The purpose of this simulation set was to see the effect of elevated concentrations of stabilized small clusters on the growth of the cluster population.

Finally, we also performed an additional set of simulations by varying monomer source rate and saturation vapor pressure

simultaneously. The monomer source rate was varied between $1 \times 10^3$ cm$^{-3}$ s$^{-1}$ and $5 \times 10^4$ cm$^{-3}$ s$^{-1}$ and the saturation vapor pressure between $1 \times 10^{-9}$ Pa and $1.5 \times 10^{-10}$ Pa in different simulations. The loss coefficient was set to $10^{-3}$ s$^{-1}$. A Gibbs free energy profile containing a maximum and no minima was assumed.

It should be noted that the studied ranges of different parameters were selected so that our analysis methods were valid in the simulated conditions. If the monomer concentration was set to a too low value, or the saturation vapor pressure and loss factor



were too high, the concentration of clusters in the largest size bins would not increase in the simulation and determining growth rates would not be reasonable. On the other hand, if the monomer concentration was very high, or the saturation vapor pressure and loss coefficient very low, a significant fraction of the flux from a certain size bin would end up not only in the next size bin but also in the size bins larger than that. In this case, the method that we use to calculate the flux-equivalent growth rate would not be valid.

## 3 Results and discussion

We determined the collision-evaporation fluxes between different size bins ($J_{\text{true}}$) and the fluxes calculated from the appearance times ($J_{\text{app}}$; see Eq. (5)) from all the simulations. We also calculated how large fraction of the true flux $J_{\text{true}}$ from each size bin is due to the collision and evaporation processes involving two clusters compared to the total flux including also monomer collisions and evaporations; hereinafter this is referred to as the non-monomer fraction of the flux. Then, we determined different growth rates (AGR, FGR and CGR) based on Eqs. (3), (4) and (7) for all the size bins. The growth rates were determined both with respect to the change in the number of molecules of cluster (denoted with the subscript N) and the change in the cluster mass diameter (denoted with the subscript D). Therefore, the figures presenting the size dependency of the growth rates are shown using two different definitions for the bin size. The ratios of the different growth rates and the fluxes from different size bins are shown as a function of the number of molecules, with the corresponding diameters presented on the upper $x$-axis. In all the figures, the fluxes originating from a certain size bin and the growth rates of that bin are plotted at the upper limit of the size bin. $J_{\text{true}}$, FGR and CGR were determined both at the mean appearance times of the clusters in each bin and at the final steady state. The figures are, though, presented only for the appearance time case, the results for the steady state being qualitatively similar.

Sections 3.1 and 3.2 discuss the results of the simulations where the free energy profile was assumed to have a single maximum and no minima, and the effects of the monomer concentration (3.1), the saturation vapor pressure, i.e. cluster stability (3.2), and the magnitude of the external sink (3.2) on the fluxes and growth rates were studied. Section 3.3 focuses on the effect of the width of the size bins on the results. Finally, Sect. 3.4 presents the results of the simulations where a different free energy profile, leading to elevated concentrations of small stable clusters, was used.

### 3.1 Effect of monomer concentration

In the first simulation set, the steady-state monomer concentration was varied to see how it affects the growth of the cluster population. When the monomer source rate, and thus also steady-state monomer concentration, increases, the non-monomer fraction of the flux becomes higher (Fig. 1a) as the relative number of clusters compared to monomers increases. At $C_{mon} = 10^6 \, \text{cm}^{-3}$ and at $C_{mon} = 5 \times 10^6 \, \text{cm}^{-3}$, the non-monomer fraction ranges from 2% to 9% and from 10% to 18%, and is highest in the smallest size bin. At $C_{\text{mon}} = 10^7 \, \text{cm}^{-3}$, the non-monomer fraction varies between 28% and 40% and is highest in the size bin





of 21–30-mers. The observed size-dependency of the non-monomer fraction of the flux is likely mainly due to the size-dependency of cluster concentrations and their losses.

In Fig. 1b the true collision-evaporation fluxes from each size bin ($J_{true}$; solid line) and the fluxes calculated from the appearance times ($J_{app}$; dashed line) are presented. Both $J_{true}$ and $J_{app}$ increase with the increasing steady-state monomer concentration.

This is due to higher cluster concentrations and, in the case of $J_{app}$, shorter time between the appearances of adjacent clusters ($\Delta t_{app}$). Furthermore, $J_{app}$ and $J_{true}$ decrease with increasing cluster size because of the decreasing cluster concentrations, with the most prominent decrease observed for the lowest monomer concentration. At a low vapor concentration the relative role of the external sink becomes more significant, and therefore, the relative decrease in the cluster concentrations accumulates more strongly with increasing cluster size.

Figures 1c and 1d present the different growth rates as a function of the number of molecules in the cluster and the cluster diameter. FGR is shown as solid lines, AGR as dashed lines, and CGR as dotted lines. All growth rates are generally higher when the steady-state monomer concentration is higher. This is due to higher values of fluxes in the case of FGR and shorter time between the appearances of adjacent size bins ($\Delta t_{app}$) in the case of AGR. For CGR the dependency on the monomer concentration follows directly from Eq. (7). From Fig. 1c we can also see that $FGR_N$ increases with the number of molecules

in the cluster, which is caused by a relatively stronger decrease in the concentration of clusters with size compared to the decrease in the fluxes (see Eq. (3)). $AGR_N$ also generally increases with the number of molecules in the cluster due to decreasing $\Delta t_{app}$, but has a minimum in the size bin of 21–30-mers at the lowest monomer concentration. As explained by Olenius et al. (2014) this may be caused by the time evolution of the evaporation fluxes from large clusters to small clusters: these fluxes are lowest at the appearance times of the clusters in the smallest size bin, which may increase the AGR of the

small sizes. $CGR_N$ also increases with the number of molecules in the cluster. Furthermore, Fig. 1d shows that $FGR_D$ increases with the cluster diameter, although the increase is very weak in the simulation with the lowest monomer concentration. On the other hand, $AGR_D$ decreases with the cluster diameter because the change in the diameter as a result of the addition of one molecule becomes smaller with the increasing cluster size. Finally, $CGR_D$ decreases with the cluster diameter according to Eq. (7).

We also studied the ratio of AGR to FGR (solid line in Fig. 1e), and the ratios of CGR to FGR (dotted line in Fig. 1e) and AGR to CGR (Fig 1f). AGR is higher than FGR at all sizes, their ratio depending strongly on the steady-state monomer concentration and the size bin. The AGR to FGR ratio generally increases with decreasing monomer concentration, reaching the highest values at $C_{mon} = 10^6$ cm$^{-3}$. However, at the largest size bins the ratio is slightly lower at $C_{mon} = 5 \times 10^6$ cm$^{-3}$ than at $C_{mon} = 10^7$ cm$^{-3}$, which is due to the size-dependency of the external sink. The AGR to FGR ratio is highest at the smallest size bin (~$10^2$–

$10^{10}$ depending on the monomer concentration) and lowest at the largest size bin (~1.4–4). The CGR to FGR ratio behaves generally in a similar way as the AGR to FGR ratio, being slightly closer to one at the highest monomer concentration. Thus, it seems that FGR of the smallest clusters can be significantly lower than AGR and CGR, especially at low monomer concentrations. This is caused by the fact that when calculating FGR, the flux from the size bin is divided by the mean value of the size distribution function in that bin ($C_i/\Delta D_{p,i}$ in Eq. (3)), while, theoretically, it should be the value at the bin boundary




(see also Vuollekoski et al., 2012). This assumption affects the results most clearly in the smallest size bin, where the concentration decreases very fast as the function of the cluster size (see Fig A3 in Appendix) and the largest contribution to the total concentration comes from the vapor monomer. For this same reason, using Eq. (5) to calculate $J_{app}$ from AGR often results in too high values compared to the real particle flux. On the other hand, AGR and CGR are close to each other at all

sizes; their ratio ranges from 0.8 to 4 at $C_{mon} = 10^6$ cm$^{-3}$, from 0.7 to 1.0 at $C_{mon} = 5 \times 10^6$ cm$^{-3}$, and from 0.9 to 1.8 at $C_{mon} = 10^7$ cm$^{-3}$ (Fig. 1f). The similarity of AGR and CGR is rather surprising when considering the very different definitions of these growth rates (see Sect. 2.1), and the fact that AGR is affected by all possible collision and evaporation processes between the clusters, while CGR is derived considering only the condensation of single molecules.

## 3.2 Effect of saturation vapor pressure and external sink

In the second simulations set the effect of cluster evaporation rate was studied by varying the saturation vapor pressure. When the saturation vapor pressure is lowered from $1 \times 10^{-9}$ Pa to $1.5 \times 10^{-10}$ Pa, the non-monomer fraction of the flux from the smallest size bin increases from 7% to 23% (Fig. 2a). In the largest size bin the non-monomer fraction varies between 2% and 15%. This shows that if the saturation vapor pressure is low, and therefore evaporation fluxes small, the cluster concentrations may rise so high that non-monomer collisions have a considerable effect on the growth of a cluster population.

The collision-evaporation fluxes for all size bins ($J_{true}$; solid line in Fig. 2b) are higher when the saturation vapor pressure is low, which is due to higher cluster concentrations. Similarly, the flux derived from appearance times ($J_{app}$; dashed line in Fig. 2b) generally increases with decreasing saturation vapor pressure, which is due to higher concentrations and shorter time between the appearance times of different size bins ($\Delta t_{app}$).

The flux-equivalent growth rate FGR increases when saturation vapor pressure is lowered because of larger fluxes (Figs 2c

and 2d). Except for the smallest size bin, AGR is also higher with the lower saturation vapor pressures due to shorter $\Delta t_{app}$. In the smallest size bin AGR is highest when the saturation vapor pressure is highest, because the small clusters reach their appearance time faster in this case. CGR is also slightly higher when the saturation vapor pressure is lower. This may seem illogical as CGR depends only on the monomer concentration, which is the same at the appearance time of the monomer in all the simulations. However, similar to FGR, CGR is determined for each size bin at the mean appearance time of that bin, and

not at the appearance time of the monomer. Thus, the differences in CGR are caused by differences in the appearance times of the size bins with varying saturation vapor pressures. Figure 2c also shows that $FGR_N$ and $CGR_N$ increase with the number of molecules in the cluster with all the saturation vapor pressures. $AGR_N$ increases as a function of size with the lower saturation vapor pressures of $2 \times 10^{-10}$ Pa and $1.5 \times 10^{-10}$ Pa, but has a minimum in the size bin of 21–30-mers when saturation vapor pressure is $1 \times 10^{-9}$ Pa. This may result from the time development of the evaporation fluxes, as discussed in Sect. 3.1. $FGR_D$ increases

with the cluster diameter, and $AGR_D$ and $CGR_D$ decrease with the cluster diameter, regardless of the saturation vapor pressure (Fig. 2d).

The ratios of AGR and FGR (solid line in Fig. 2e) and CGR and FGR (dotted line in Fig. 2e) depend strongly on the saturation vapor pressure and the size bin. Still, with all three saturation vapor pressures AGR and CGR are higher than FGR at all sizes.



In the smallest size bin, the AGR to FGR ratio varies between $10^3$ and $10^9$ increasing with the saturation vapor pressure. The CGR to FGR ratio behaves similarly as the AGR to FGR ratio, but it is slightly higher at the largest sizes when $p_{sat} = 1 \times 10^{-9}$ Pa. Altogether, FGR gives clearly lower growth rates than AGR and CGR for the smallest clusters, with the differences increasing when the saturation vapor pressure, and thus also evaporation fluxes, become larger. However, in the largest size bin AGR and CGR are close to FGR: the AGR to FGR ratio ranges from 1.4 to 1.7 and the CGR to FGR ratio from 1.3 to 2.7. Furthermore, AGR and CGR are close to each other with all the saturation vapor pressures (Fig. 2f). When $p_{sat} = 1 \times 10^{-9}$ Pa, the AGR to CGR ratio varies between 0.5 and 1.3, being highest at the smallest size bin. With $p_{sat} = 2 \times 10^{-10}$ Pa and $p_{sat} = 1.5 \times 10^{-10}$ Pa, the AGR to CGR ratio ranges from 0.7 to 1.0 and from 0.8 to 1.2, increasing with increasing cluster size.

In the third simulation set, the effect of the external sink on the growth of cluster population was studied by varying the value of the external monomer and cluster sink from $0.7 \times 10^{-3}$ s$^{-1}$ to $2 \times 10^{-3}$ s$^{-1}$. Lowering the loss coefficient seems to have similar effects on the results as lowering the saturation vapor pressure (see Fig. A2 in Appendix). This results from the fact that in both cases the number of clusters increases, and therefore the cluster collisions become more important relative to evaporation or other losses. When the loss coefficient is lowered, the non-monomer fraction of the flux increases, $J_{true}$ and $J_{app}$ get higher values, and FGR and CGR increase, as expected. AGR also increases, except for the smallest size bin, where AGR is higher with a higher loss coefficient due to shorter $\Delta t_{app}$. The AGR to FGR ratio increases with the loss coefficient: for instance, in the smallest size bin the AGR to FGR ratio ranges from 700 to $10^6$, and in the largest size bin the ratio is between 1.4 and 2.2. The AGR to CGR ratio, on the other hand, varies from 0.6 to 1.7 in the smallest size bin and from 1.0 to 1.2 in the largest size bin. The highest values of the ratio are obtained with the highest loss coefficient.

### 3.3 Effect of size resolution

In the fourth simulation set, the width of size bins was varied too see how the size resolution affects the growth rates. When the size bins are wider, the non-monomer fraction of the flux at a certain size is higher (Fig. 3a). This is partly due to the size dependency of the non-monomer fraction, and partly due to the differences in the appearance times of bins with different widths, as the values are determined at the appearance times.

The collision-evaporation fluxes ($J_{true}$; solid line in Fig. 3b) are not greatly affected by the size bin width as the flux from the size bin originates mostly from the largest clusters of that bin. Therefore, the small differences in $J_{true}$ obtained with different size bin widths are mainly caused by the differences in the appearance times of bins with different widths. On the other hand, the flux calculated from the appearance times ($J_{app}$; dashed line in Fig. 3b) at a certain size becomes lower when the size bin width is decreased. This results from the decrease of the mean value of the size distribution function of the bin ($C_i/\Delta D_{p,i}$ in Eq. (5)) used for calculating $J_{app}$.

The flux-equivalent growth rate FGR at a certain size, on the other hand, increases when the bin width is decreased (Figs 3c and 3d), due to the lower mean value of the size distribution function of the bin ($C_i/\Delta D_{p,i}$ in Eq. (3)). Also, CGR becomes slightly higher when the bin width is decreased. Furthermore, AGR is also higher with narrower size bins as then the size bin width is relatively higher compared to $\Delta t_{app}$ than with wider size bins (see Eq. (4)).





The relation of different growth rates to each other is also affected by the width of the size bins (Figs 3e and 3f). The AGR to FGR ratio gets higher values when the bin sizes are wider. In the smallest size bin the ratio is $10^3$–$10^5$ depending on the bin width, and in the largest size bin the ratio is correspondingly 1.6–5.4. The CGR to FGR ratio is slightly lower than the AGR to FGR ratio with all the bin widths. The ratio of AGR to CGR, on the other hand, slightly increases with decreasing bin size. In the smallest size bin the ratio is 0.9–1.4 depending on the bin width, while in the largest size bin the ratio is ~1.1 in all cases. Altogether, high size resolution seems beneficial when using FGR to describe the growth of the cluster population, or when calculating particle fluxes from growth rates utilizing Eq. (5).

In order to compare our results directly with those of Olenius et al. (2014), we performed additional simulations with the saturation vapor pressure of $1 \times 10^{-9}$ Pa and allowing only monomer collisions and evaporation in our system. In this case the AGR to FGR ratios with different bin widths become higher compared to the simulations where cluster collisions contribute to the growth. The CGR to AGR ratio does not change as significantly.

## 3.4 Effect of stable small clusters

In the fifth simulation set a different cluster free energy profile was used to study the effect of elevated concentrations of stable small clusters on the growth of the population. The contribution of non-monomer collisions to the fluxes between different size bins is significantly increased by the stabilization of small clusters (Fig. 4a, see also Fig. 1a for a comparison). In the smallest size bin the growth mainly proceeds by non-monomer collisions: the non-monomer fraction of the flux is 56–71% with different monomer concentrations. In the largest size bin, the non-monomer fraction depends strongly on the steady-state monomer concentration: the fraction is 15% with the lowest monomer concentration and 62% with the highest monomer concentration.

The collision-evaporation fluxes ($J_{true}$; solid line in Fig. 4b) and the fluxes derived from the appearance times of clusters ($J_{app}$; dashed line in Fig. 4b) also increase in the presence of stabilized small clusters (see Fig. 1b for a comparison). Correspondingly, FGR and AGR are higher, while CGR does not change significantly (Figs 4c and 4d, see Figs 1c and 1d for a comparison). The ratios of AGR to FGR (solid line in Fig. 4e) and CGR to FGR (dotted line in Fig. 4e) are lower when there are small stable clusters present (see Fig. 1e for a comparison). This is clear especially at small sizes, indicating that FGR increases there more than AGR or CGR due to the elevated concentrations of small clusters. The increase of FGR in the smallest size bin can be explained by a slower decrease of the concentration as a function of the cluster size in the presence of small stable clusters (see Fig A3 in Appendix). The AGR to FGR ratio varies between $10^2$ and $10^8$ in the smallest size bin and between 1.5 and 3.5 in the largest size bin, being highest with the lowest monomer concentration. The CGR to FGR behaves similarly to the AGR to FGR ratio; the most notable difference is that the CGR to FGR ratio is below one (0.6–0.7) at the largest sizes when $C_{mon} = 5 \times 10^6$ cm$^{-3}$ and $C_{mon} = 10^7$ cm$^{-3}$. On the other hand, the presence of small stable clusters increases the AGR to CGR ratio slightly (Fig. 4f, see Fig. 1f for a comparison). The AGR to CGR ratio varies between 0.8 and 6 at $C_{mon} = 10^6$ cm$^{-3}$, between 1.2 and 2.0 at $C_{mon} = 5 \times 10^6$ cm$^{-3}$, and 1.8 and 4.2 at $C_{mon} = 10^7$ cm$^{-3}$.





### 3.5 Combined effect of external conditions and the properties of model substance

To see the combined effect of external conditions and the properties of model substance on the growth of clusters, an additional set of simulations was performed by varying monomer source rate and saturation vapor pressure simultaneously. A Gibbs free energy profile containing a maximum and no minima was assumed. Figure 5a shows the non-monomer fraction of the flux from the smallest size bin (solid lines) and from the largest size bin (dashed lines) in all these simulations. The ratio of the monomer source rate to the loss coefficient ($Q/S$), which largely determines how the system behaves, is presented on the *x*-axis, and the color of the line shows the saturation vapor pressure. The non-monomer fraction of the flux increases with increasing $Q/S$ and decreasing saturation vapor pressure. For the highest saturation vapor pressure, the non-monomer fraction ranges from 6% to 21% in the smallest size bin and from 1% to 17% in the largest size bin, while for the lowest saturation vapor pressure, the ratios are 10%–53% and 3%–44% in the smallest and largest bin, respectively.

The ratio of AGR to FGR and AGR to CGR in different simulations is presented for the smallest size bin (solid lines) and the largest size bin (dashed lines) in Figs 5b and 5c. In the smallest size bin, the AGR to FGR ratio decreases with increasing $Q/S$ and decreasing saturation vapor pressure. In the largest size bin, however, the ratio has a minimum at $Q/S = 5 \times 10^6$ cm$^{-3}$ with the lowest saturation vapor pressures, and at highest $Q/S$-values the ratio is lowest when $p_{sat} = 10^{-9}$ Pa. In the smallest size bin the AGR to FGR ratio ranges from 500 to $10^{16}$ and from 100 to $10^9$ with the highest and lowest saturation vapor pressures, respectively; in the largest size bin the corresponding ranges for the ratio are 1.4–$10^4$ and 2.2–3.3. These results show that, depending on the external conditions and the properties of model substance, FGR can be clearly lower than AGR, especially at the smallest sizes. This is related to the fact that in the smallest size bin FGR is not the best quantity for describing the cluster growth rate (see the last paragraph of Sect. 3.1). On the other hand, if one wishes to use the growth rate to estimate the particle fluxes $J_{app}$ calculated from AGR using Eq. (5) are in these cases significantly too high. AGR and CGR are considerably closer to each other compared to FGR and AGR or CGR. In the smallest size bin the AGR to CGR ratio varies from 0.5 to 9.5 and from 1.3 to 3.3 with the highest and lowest saturation vapor pressures, respectively, and in the largest size bins the corresponding ranges for the ratios are 1.3–1.6 and 0.8–3.

Finally, we also studied the total concentration of clusters (2–70-mers) in different simulations. Figure 5d shows that the concentration of clusters increases with increasing $Q/S$ and decreasing saturation vapor pressure. With the highest saturation vapor pressure the steady state cluster concentration varies from $1.9 \times 10^3$ cm$^{-3}$ to $2.9 \times 10^6$ cm$^{-3}$, and with the lowest saturation vapor pressure from $1.2 \times 10^4$ cm$^{-3}$ to $4.3 \times 10^6$ cm$^{-3}$. When comparing Figs 5a–5d, we may conclude that in the conditions where the concentration of clusters becomes high, and thus their collisions become more important relative to evaporation and other losses, the contribution of non-monomer collisions to the growth of clusters becomes significant. Furthermore, in these conditions growth rates determined with different methods tend to be closer to each other than in the conditions where cluster concentrations are lower.




## 4 Conclusions

We used a dynamic model to simulate the time evolution of cluster concentrations in a system where cluster-cluster collisions significantly contribute to the growth of clusters. More specifically, we studied how consistent the flux-equivalent growth rate (FGR), the growth rate derived from the appearance times of the clusters (AGR), and the growth rate calculated based on

irreversible vapor condensation (CGR) are with each other in different, atmospherically relevant, conditions.

In majority of the simulations the Gibbs free energy of formation of the clusters was assumed to have a single maximum and no minima, which corresponds to the increasing stability of clusters with increasing cluster size. In most of the simulations FGR was lower than AGR and CGR. The difference was highest, often several orders of magnitudes, in the smallest size bin (at ~1.2 nm). This results from the very low values of FGR at the smallest sizes, caused by the approximations made in its

derivation. In the largest size bin (at ~2.2 nm), FGR was closer to AGR and CGR. The difference between FGR and AGR or CGR was observed to decrease in conditions where cluster concentrations are high and thus evaporation and other losses are less important, i.e. when the monomer source rate is high, when the saturation vapor pressure is low and when the external losses of clusters are low. Furthermore, in these conditions a higher fraction of the flux was found to be due to cluster-cluster collisions than in the conditions with lower cluster concentrations. Finally, it was observed that AGR and CGR are clearly

closer to each other than to FGR; their difference was often very small, and within the factor of 10 in all the simulations. This is rather surprising since AGR is affected by all possible collision and evaporation processes between the clusters, while CGR is derived considering only the condensation of single molecules.

In one simulation set, a different free energy profile was used, leading to elevated concentrations of stable small clusters, which could correspond to the situation in the atmosphere. In this case, a significantly higher fraction of the growth was due to cluster-

cluster collisions than in other simulations. Furthermore, the growth rates of clusters were higher and the different growth rates were closer to each other than in the simulations without stable small clusters.

Moreover, the used size resolution, i.e. the size bin width, was observed to affect the relation between the different growth rates. Generally, the difference between the different growth rates increased with increasing size bin width. Thus, when determining growth rates from measured particle size distributions, a size resolution as high as possible should be used.

Altogether, our results demonstrate that different approaches to determine the growth rates of nanometer-sized clusters may give different values depending on the ambient conditions, the properties of the condensing vapor and the clusters, and the size resolution used in the analysis. Especially at the smallest, sub-2nm sizes, the differences between growth rates deduced with different methods can be significant. The results also indicate that the conventional method used to determine particle formation rates from growth rates, may give estimates far from their true values. This should be kept in mind when applying

these methods to measured particle size distributions, and utilizing the results in particle formation event analyses.

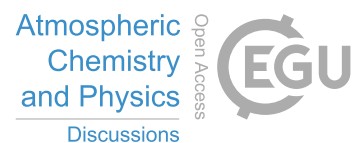

## Acknowledgements

This research was supported by the The Academy of Finland Centre of Excellence program (grant no. 272041), the European Research Council (ERC) projects ATM-NUCLE (grant no. 227463), MOCAPAF (grant no. 257360), and ATMOGAIN (grant no. 278277), and the European Union's Horizon 2020 research and innovation program under the Marie Sklodowska-Curie

(grant no. 656994).

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





Table 1. Summary of the performed simulations.

| Simulation set 1: Varying monomer source rate | | | |
|---|---|---|---|
| Monomer concentration (cm$^{-3}$) | $1\times10^6$ | $5\times10^6$ | $1\times10^7$ |
| Loss coefficient (s$^{-1}$) | $10^{-3}$ s$^{-1}$ | $10^{-3}$ s$^{-1}$ | $10^{-3}$ s$^{-1}$ |
| Saturation vapor pressure (Pa) | $2\times10^{-10}$ | $2\times10^{-10}$ | $2\times10^{-10}$ |
| Size bin width (molecules) | 10 | 10 | 10 |
| *Simulation set 2: Varying saturation vapor pressure* | | | |
| Monomer concentration (cm$^{-3}$) | $5\times10^6$ | $5\times10^6$ | $5\times10^6$ |
| Loss coefficient (s$^{-1}$) | $10^{-3}$ s$^{-1}$ | $10^{-3}$ s$^{-1}$ | $10^{-3}$ s$^{-1}$ |
| Saturation vapor pressure (Pa) | $1.5\times10^{-10}$ | $2\times10^{-10}$ | $1\times10^{-9}$ |
| Size bin width (molecules) | 10 | 10 | 10 |
| *Simulation set 3: Varying loss coefficient* | | | |
| Monomer concentration (cm$^{-3}$)* | $6.4\times10^6$ | $5\times10^6$ | $2.6\times10^6$ |
| Loss coefficient (s$^{-1}$) | $7\times10^{-4}$ s$^{-1}$ | $10^{-3}$ s$^{-1}$ | $2\times10^{-3}$ s$^{-1}$ |
| Saturation vapor pressure (Pa) | $2\times10^{-10}$ | $2\times10^{-10}$ | $2\times10^{-10}$ |
| Size bin width (molecules) | 10 | 10 | 10 |
| *Simulation set 4: Varying size bin width* | | | |
| Monomer concentration (cm$^{-3}$) | $5\times10^6$ | $5\times10^6$ | $5\times10^6$ |
| Loss coefficient (s$^{-1}$) | $10^{-3}$ s$^{-1}$ | $10^{-3}$ s$^{-1}$ | $10^{-3}$ s$^{-1}$ |
| Saturation vapor pressure (Pa) | $2\times10^{-10}$ | $2\times10^{-10}$ | $2\times10^{-10}$ |
| Size bin width (molecules) | 5 | 10 | 14 |
| *Simulation set 5: Varying monomer source rate with a different Gibbs free energy profile* | | | |
| Monomer concentration (cm$^{-3}$) | $1\times10^6$ | $5\times10^6$ | $1\times10^7$ |
| Loss coefficient (s$^{-1}$) | $10^{-3}$ s$^{-1}$ | $10^{-3}$ s$^{-1}$ | $10^{-3}$ s$^{-1}$ |
| Saturation vapor pressure (Pa) | $2\times10^{-10}$ | $2\times10^{-10}$ | $2\times10^{-10}$ |
| Size bin width (molecules) | 10 | 10 | 10 |

* The monomer source rate was set to $5.5\times10^3$ cm$^{-3}$ s$^{-1}$ in all these simulations.





Figure 1. The effect of steady-state monomer concentration (shown as different colors) on quantities describing cluster growth: a) non-monomer fraction of flux from each size bin, b) the true collision-evaporation flux from each size bin ($J_{true}$; solid line) and the fluxes calculated from the appearance times of clusters ($J_{app}$; dashed line), c) and d) flux-equivalent growth rate (FGR; solid line), growth rate derived from the appearance times of clusters (AGR; dashed line), and growth rate calculated based on irreversible vapor condensation (CGR; dotted line), e) the ratios of AGR to FGR (solid line) and CGR to FGR (dotted line), f) the ratio of AGR to CGR.





Figure 2. The effect of saturation vapor pressure (shown as different colors) on quantities describing cluster growth: a) non-monomer fraction of flux from each size bin, b) the true collision-evaporation flux from each size bin ($J_{true}$; solid line) and the fluxes calculated from the appearance times of clusters ($J_{app}$; dashed line), c) and d) flux-equivalent growth rate (FGR; solid line), growth rate derived from the appearance times of clusters (AGR; dashed line), and growth rate calculated based on irreversible vapor condensation (CGR; dotted line), e) the ratios of AGR to FGR (solid line) and CGR to FGR (dotted line), f) the ratio of AGR to CGR.





Figure 3. The effect of size bin width (shown as different colors) on quantities describing cluster growth: a) non-monomer fraction of flux from each size bin, b) the true collision-evaporation flux from each size bin ($J_{true}$; solid line) and the fluxes calculated from the appearance times of clusters ($J_{app}$; dashed line), c) and d) flux-equivalent growth rate (FGR; solid line), growth rate derived from the appearance times of clusters (AGR; dashed line), and growth rate calculated based on irreversible vapor condensation (CGR; dotted line), e) the ratios of AGR to FGR (solid line) and CGR to FGR (dotted line), f) the ratio of AGR to CGR.





Figure 4. The effect of steady-state monomer concentration (shown as different colors) on quantities describing cluster growth *in the presence of stable small clusters:* : a) non-monomer fraction of flux from each size bin, b) the true collision-evaporation flux from each size bin ($J_{true}$; solid line) and the fluxes calculated from the appearance times of clusters ($J_{app}$; dashed line), c) and d) flux-equivalent growth rate (FGR; solid line), growth rate derived from the appearance times of clusters (AGR; dashed line), and growth rate calculated based on irreversible vapor condensation (CGR; dotted line), e) the ratios of AGR to FGR (solid line) and CGR to FGR (dotted line), f) the ratio of AGR to CGR.





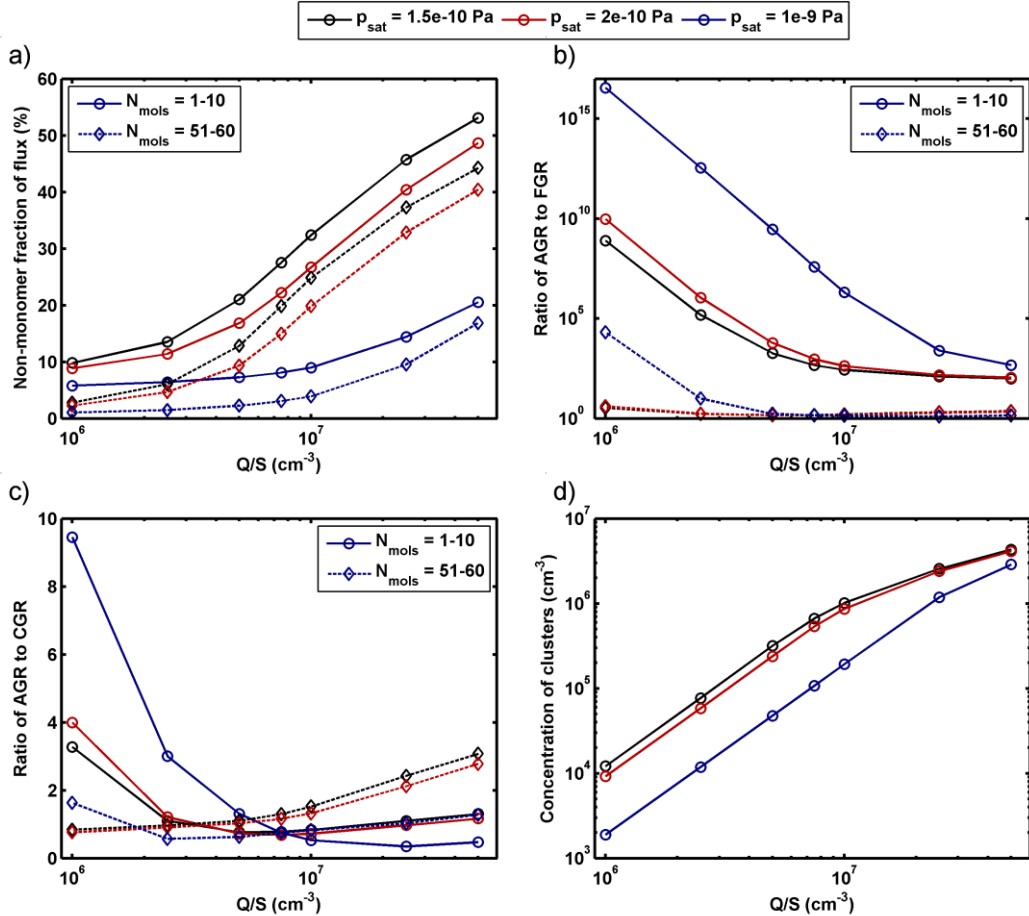

Figure 5. The results of the simulations with different saturation vapor pressures ($p_{sat}$; shown as different colors) and different ratios of monomer source rate to the loss coefficient ($Q/S$; shown on the *x*-axis) in the smallest size bin (solid line) and in the largest size bin (dashed line): a) non-monomer fraction of the flux, b) the ratio of appearance time growth rate (AGR) to flux-equivalent growth rate (FGR), c) the ratio of appearance time growth rate to growth rate calculated based on irreversible vapor condensation (CGR), d) steady state concentration of all clusters (2–70-mers).



**Appendix**

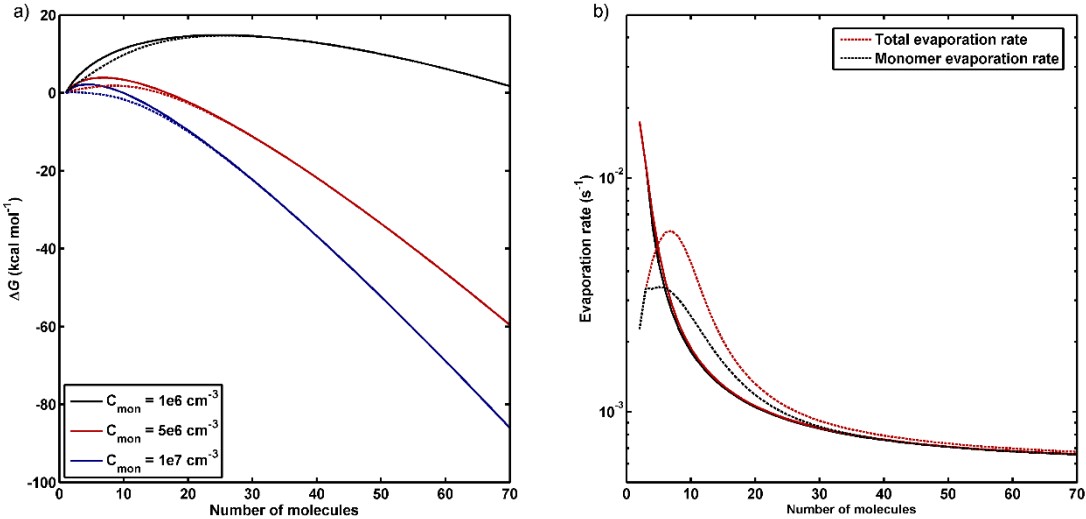

Figure A1. a) The classical Gibbs free energy profile with one maximum and no minima (solid lines) and with an additional negative term corresponding to stabilization of the smallest clusters (dashed lines) at different steady state monomer concentrations (shown as different colors). b) Evaporation profiles corresponding to the two Gibbs free energy profiles (note that the evaporation rate is independent of the vapor concentration).



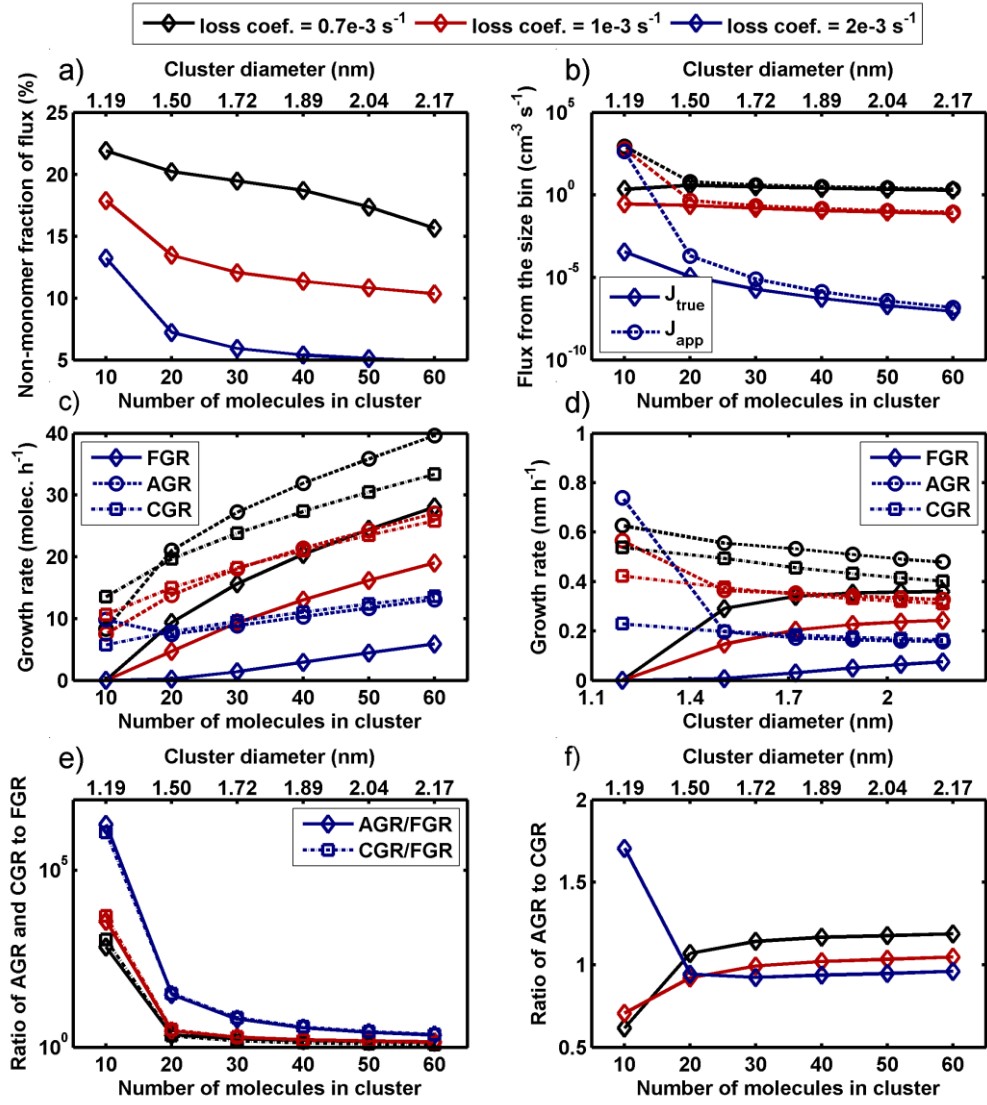

Figure A2. The effect of loss coefficient (shown as different colors) on quantities describing cluster growth in the presence small stable clusters: a) non-monomer fraction of flux from each size bin, b) collision-evaporation flux from each size bin ($J_{true}$; solid line) and the fluxes calculated from the appearance times of clusters ($J_{app}$; dashed line), c) and d) flux-equivalent growth rate (FGR; solid line), growth rate derived from the appearance times of clusters (AGR; dashed line), and growth rate calculated based on irreversible vapor condensation (CGR; dotted line), e) the ratios of AGR to FGR (solid line) and CGR to FGR (dashed line), f) the ratio of AGR to CGR.



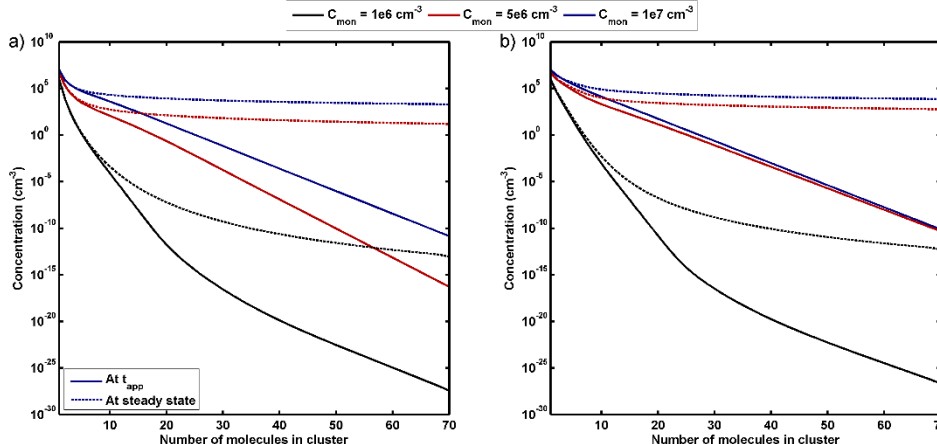

Figure A3. Cluster distribution at the mean appearance time of the two smallest size bins (solid lines) and at the final steady state (dashed lines) when a) the Gibbs free energy profile has one maximum and no minima b) the Gibbs free energy profile has an additional negative term corresponding to the stabilization of the smallest clusters. The colors show the steady state monomer concentration.