# Peer review of "Growth of atmospheric clusters involving cluster-cluster collisions: comparison of different growth rate methods"

_Atmospheric Chemistry and Physics, 2015_

## Referee Comment (RC1) · Anonymous Referee #1 · 20 Feb 2016

The paper explores the differences in estimated growth rates for sub-3nm particles by three different methods. The work seems solid and thorough, and it is of publishable quality. However, I'm unsure if the interest is broad enough for ACP as opposed to e.g. Aerosol Science & Technology. I'd recommend either GMD or AMT, though the paper is a modelling study that is of interest to observationalists, so it sits between those journals. I'll leave it up to the editor to make a judgement call.

If the paper is to be published in ACP, I have several minor comments that should be addressed.

P2 L11: "CGR" doesn't correspond to the previous text. Can you define it? Is it "condensational growth rate"?

P3 L6-7: "AGR" and "FGR", same as above.

P4 L7-8 and throughout: I'm somewhat confused by the different sinks in this paper. Is the loss due to coagulation with larger particles included here? Is the loss due to coagulation between the model-resolved clusters included elsewhere when discussing the "loss coefficient"?

Eqn 1: What about the source by a collision of two clusters that are not in the previous bin, but where the sum of their molecules would put the resultant particle in the current bin?

Sect 2.2: Would it have made sense to do a simulation set where you simulation every cluster size from 1-70, so you don't have any numerical issues? this would be a useful comparison.

P7 L4: Why is the concentration in units cm-3 s-1?

P7 L30: What values of monomer sources were used earlier (I think only steady state concentrations were given)?

---

## Referee Comment (RC2) · Anonymous Referee #2 · 24 Feb 2016

In their manuscript, Kontkanen et al. present a set of numerical simulations of a one-component system, with the purpose to study the effect of cluster-cluster collisions on the growth of particles, and also the impact of different definitions and methods of the growth rate. The study is interesting, and gives a good overview of the different ways of looking at fresh particle growth, and the author's approach brings interesting new viewpoints to recent studies of nanoparticle formation in the atmosphere. The work is quite theoretical, and no comparison to actual experiments is made (as the studied one-component system would be quite difficult to produce). I think that the author's work is useful and interesting, and also in the scope of ACP.

In my opinion the manuscript could be published in ACP, but I would appreciate it if

some more information on the simulations and their analysis could be provided. I have listed my concerns below.

* I did not fully follow how the simulations were performed. Were all inputs kept constant over the whole time? The authors state that there was an initial cluster distribution corresponding to a low monomer concentration. I addition, it is stated that a 'steady-state monomer concentration' is reached at some point. This suggest that the aerosol formation and growth proceeds until some kind of a steady state is reached. It it is so, this has at leat the following implications for the data analysis:

a) from which part of the process are the various size-dependent growth rates determined? For AGR, the GR is necessarily from the start, but for the other GRs some other times could be used. Was the steady-state situation used? What was the criterium for steady state? Were the different growth rates determined at different time points within one simulation? Were the growth rates time-dependent within one simulation?

b) A 'steady state' in which everything stays constant is rarely seen in nature, partly because the growing particles contribute to a growth in the sink and therefore the Q/S fraction also changes. This could, for example, introduce a coupling between the formation rate and the AGR of FGR, because high formation rate causes a fast-increasing sink quickly to the distribution. I understand that the sink in the simulations was not dynamic, and therefore this effect is omitted. Do the authors have an idea how this affects the results?

In any case, it would be useful to have a description of the time-dependent developing distribution, for example using a figure. The same figure could then be used to give a graphical explanation of the derivation of the different growth rates.

* A big question in particle formation research is the size dependence of growth in particle formation. From the results in this paper (Figs 1d, 2d, 3d, and 4d), it seems that for this one-component case, AGR and CGR tend to decrease with size while FGR increases with size. Is this a result that could be considered general? Is it possible that

this changes in a true, multi-component case?

* Finally, I think that the discussion in the conclusions could be improved. In the last paragraph, (p. 14, L 23) the authors state that "conventional methods used to determine particle formation rates from growth rates may give estimates far from their true values". Based on earlier discussion, it seems that the authors bring forward the idea that not a single 'true' value exists (for example, the growth rate can mean the AGR, FGR or CGR). In the study of particle formation, finding the values for the formation rate J and the growth rate GR has for some time been a key objective. It would be useful if the authors could discuss here which of the given growth rates is the true value meant.

Specific comments:

P7 L33: Regarding the monomer concentration ranges, I understand the limitation of the lower level. However, the upper level for the monomer concentration is less justified. This is a case of selecting a too long time step; reducing the time step in the differential equation solving would present usable results. In any case, the limits for the usability of the methods should bee given.

P19, Table 1: For reproducibility, it is not enough to give the final monomer concentration, but also the monomer source rate is important information. I think that the monomer source rates used to obtain the final concentrations in Table 1 should be given also in Table 1.

---

## Author Response (AR1)

**FINAL RESPONSE**

The replies to all referee comments can be found below together with the revised manuscript. The changes made in the manuscript are marked in yellow. Note that in addition to the changes mentioned in the replies, Fig. 3 was slightly changed. This did not, however, change the results significantly.

**Reply to Referee #1**

We thank Anonymous Referee #1 for their helpful comments. We have answered to the comments below. The bold text is quoted from the referee's comments, and the text in italics has been added to the manuscript.

**GENERAL COMMENTS**

The paper explores the differences in estimated growth rates for sub-3nm particles by three different methods. The work seems solid and thorough, and it is of publishable quality. However, I'm unsure if the interest is broad enough for ACP as opposed to e.g. Aerosol Science & Technology. I'd recommend either GMD or AMT, though the paper is a modelling study that is of interest to observationalists, so it sits between those journals. I'll leave it up to the editor to make a judgement call. If the paper is to be published in ACP, I have several minor comments that should be addressed.

We believe that our paper fits into the scope of ACP (as also agreed by Referee #2) and is relevant for the readership of the journal, as the methods discussed in the paper are commonly used for analyzing experimental data on new particle formation both in the atmosphere and in chamber experiments. Experimentally determined growth rates are used to characterize particle formation and growth events, and, thus, they are routinely utilized in the analysis of particle distribution data (e.g. Kulmala et al., 2012). Moreover, growth rates extracted from measurements are often compared to those determined by theoretical or modeling approaches. Therefore, assessing the differences between different growth rate definitions is useful also for modelers, and, in general, for all researchers working with atmospheric particle data.

**SPECIFIC COMMENTS**

**P2 L11: "CGR" doesn't correspond to the previous text. Can you define it? Is it "condensational growth rate"?**

Yes, in this paper CGR is used as the abbreviation for the growth rate determined based on irreversible vapor condensation, and the letters stand for condensational growth rate. This has now been added to the text.

**P3 L6-7: "AGR" and "FGR", same as above.**

AGR is used as the abbreviation for the growth rate obtained with the appearance time method, and the letters stand for appearance time growth rate. FGR is used as the abbreviation for the growth rate corresponding to the molecular fluxes, i.e. flux-equivalent growth rate. These are now clarified in the text.

**P4 L7-8 and throughout: I'm somewhat confused by the different sinks in this paper. Is the loss due to coagulation with larger particles included here? Is the loss due to coagulation between the model-resolved clusters included elsewhere when discussing the "loss coefficient"?**

We agree that the processes included in the model are not explained very clearly. The model includes all the possible collisions between the clusters (up to the clusters consisting of 70 molecules), and

also an additional, time-independent external sink. The external sink can be thought to represent the loss of the clusters due to the coagulation onto pre-existing larger particles in the atmosphere. The losses discussed in the paper always refer to the external sink, characterized by the loss coefficient of the vapor monomer (in 1/s). The loss frequencies of the clusters are somewhat lower than that of the monomer (see Eq. (8)). The role of coagulation between the modeled clusters in the cluster growth dynamics is characterized by the non-monomer fraction of the flux.

To clarify the processes included in the model, we now added the following sentences in the beginning of the Sect. 2.2 (page 6, line 16):

We simulated the time-evolution of cluster concentrations in a one-component system using the Atmospheric Cluster Dynamics Code (ACDC; McGrath et al., 2012; Olenius et al., 2014). The model included the production of monomers, all the possible collision and evaporation processes between different clusters, and the losses of clusters due to an external sink.

**Eqn 1: What about the source by a collision of two clusters that are not in the previous bin, but where the sum of their molecules would put the resultant particle in the current bin?**

It is true that Eq. (1) considers only the flux originating from the previous  $\lim J_{i-1,i}$  although the collisions between clusters belonging to  $\lim < i-1$  may also contribute to the flux into  $\lim i$ . This contribution is in practice notable only when the concentrations of large clusters are high, or the bin widths very narrow. In the simulations of this work, all possible collisions between clusters were considered but the studied conditions were selected so that the contribution of collisions from  $\lim < i-1$  to the flux into  $\lim i$  was negligible for all size classes *i*. We added the following remark in Sect. 2.1 (page 4, line 8):

In situations with high concentrations of large clusters, the overall flux into bin i may contain the contributions of fluxes from smaller bins < i-1. This makes the analysis of the dynamics more complex and such situations are not addressed in this study.

**Sect 2.2: Would it have made sense to do a simulation set where you simulation every cluster size from 1-70, so you don't have any numerical issues? This would be a useful comparison.**

Actually, every cluster size from 1 to 70 was simulated. Only when calculating the growth rates and fluxes, the clusters were grouped into the size bins. This was done as the growth rate of a single cluster is ambiguous when clusters grow not only by monomer additions but also by cluster-cluster collisions. To make this clear, we modified a sentence in the Sect. 2.2 (page 7, line 4):

After simulating the time-evolution of the discrete cluster concentrations, the clusters were grouped into size bins containing an equal number of cluster (in most cases ten), for which fluxes and growth rates were determined.

Furthermore, we want to point out that the previous study by Olenius et al. (2014) discusses the case where clusters grow only by monomer attachments, and thus the growth rates can be unambiguously determined with the resolution of one molecule.

**P7 L4: Why is the concentration in units cm-3 s-1?**

This was a typographical error that has now been corrected. The correct unit is cm-3.

**P7 L30: What values of monomer sources were used earlier (I think only steady state concentrations were given)?**

The monomer source rates used in the different simulations are now presented in Table 1.

**Reply to Referee #2**

We thank Anonymous Referee #2 for their helpful comments. We have answered to the comments below. The bold text is quoted from the referee's comments, and the text in italics has been added to the manuscript.

**GENERAL COMMENTS**

In their manuscript, Kontkanen et al. present a set of numerical simulations of a one-component system, with the purpose to study the effect of cluster-cluster collisions on the growth of particles, and also the impact of different definitions and methods of the growth rate. The study is interesting, and gives a good overview of the different ways of looking at fresh particle growth, and the author's approach brings interesting new viewpoints to recent studies of nanoparticle formation in the atmosphere. The work is quite theoretical, and no comparison to actual experiments is made (as the studied one-component system would be quite difficult to produce). I think that the author's work is useful and interesting, and also in the scope of ACP. In my opinion the manuscript could be published in ACP, but I would appreciate it if some more information on the simulations and their analysis could be provided. I have listed my concerns below.

\* I did not fully follow how the simulations were performed. Were all inputs kept constant over the whole time? The authors state that there was an initial cluster distribution corresponding to a low monomer concentration. I addition, it is stated that a 'steady-state monomer concentration' is reached at some point. This suggest that the aerosol formation and growth proceeds until some kind of a steady state is reached. If it is so, this has at least the following implications for the data analysis:

The model inputs, i.e. the monomer source rate, the external loss, temperature and the vapor properties, were kept constant during one simulation. The initial cluster distribution, corresponding to the monomer concentration of  $5 \times 10^5$  cm-3, was used so that the simulations would correspond to the real situation in, for example, the atmosphere or a chamber experiment where the concentration of condensing vapor (e.g. sulfuric acid) is higher than zero already before the actual new particle formation event. As concluded by the referee, the simulations were continued until a steady state was reached.

**a) from which part of the process are the various size-dependent growth rates determined? For AGR, the GR is necessarily from the start, but for the other GRs some other times could be used. Was the steady-state situation used? What was the criterion for steady state? Were the different growth rates determined at different time points within one simulation? Were the growth rates time-dependent within one simulation?**

AGR was obtained by differentiating ( $t_{app}$ ,  $D_p$ )-data, where the appearance time of a size bin,  $t_{app}$ , was defined as the time at which the concentration of the bin reaches 50% of the total increase in the concentration (see Sect. 2.1.2). AGR thus yields one time-independent value for each size bin in one simulation. FGR and CGR, on the other hand, were time-dependent in the simulations, as FGR depends on the time evolution of the fluxes, and CGR on the time evolution of the monomer concentration. Therefore, both FGR and CGR changed during one simulation until the steady state was reached. We determined FGR and CGR at the mean appearance times of consecutive size bins, and at the steady state. This is already mentioned on page 8, although we now slightly modified the sentence to be more exact. As the main aim of this study was to compare the different growth rates with each other, only the values obtained at the appearance times were presented in the paper. However, the steady-state results were qualitatively similar. The criterion for the steady state was that

the relative changes in the concentrations of the monomer and clusters as a function of time were negligible (less than  $10^{-4}$  %) when the integration was continued for several minutes.

b) A 'steady state' in which everything stays constant is rarely seen in nature, partly because the growing particles contribute to a growth in the sink and therefore the Q/S fraction also changes. This could, for example, introduce a coupling between the formation rate and the AGR of FGR, because high formation rate causes a fast-increasing sink quickly to the distribution. I understand that the sink in the simulations was not dynamic, and therefore this effect is omitted. Do the authors have an idea how this affects the results? In any case, it would be useful to have a description of the time-dependent developing distribution, for example using a figure. The same figure could then be used to give a graphical explanation of the derivation of the different growth rates.

The external sink, corresponding to the loss of clusters due to coagulation onto large particles in the atmosphere, was indeed kept constant in our simulations. The sink of the vapor monomer and small clusters due to collisions with the simulated larger clusters (up to the clusters containing 70 molecules), naturally increased during the simulation when the cluster concentrations were increasing. On the other hand, the clusters growing beyond the size of 70 molecules were removed from the simulation, and therefore did not contribute to the sink. This means that in our simulations the sink remained lower than if the sink due to the particles larger than 70 molecules was taken into account, and thus the concentrations of vapor monomer and clusters could increase higher. However, as we are here interested only in the beginning of a new particle formation event (when sub-3nm clusters are formed), this effect is likely not very big, as it takes several hours before the nanometer sized clusters can reach sizes large enough to contribute significantly to the total sink. Thus, the coupling between the particle formation and the sink is likely to become significant mainly in very rapid particle formation bursts, where both the initial formation and the growth occur fast. Following the referee's suggestion, we now added a figure showing the time evolution of the cluster distribution in one simulation into the Appendix (Fig. A2 in the revised manuscript).

**\* A big question in particle formation research is the size dependence of growth in particle formation. From the results in this paper (Figs 1d, 2d, 3d, and 4d), it seems that for this one-component case, AGR and CGR tend to decrease with size while FGR increases with size. Is this a result that could be considered general? Is it possible that this changes in a true, multi-component case?**

CGR decreases in all cases with the increasing cluster diameter, which results directly from its definition (see Eq. (7)). The size dependencies of AGR and FGR, on the other hand, are not affected only by the substance properties, but also by the ambient conditions, as well as the size classification, i.e. the resolution of the analysis (see Eqs (3) and (4)). The results for a multi-component system with an evaporation profile similar to our one-component substance can be expected to qualitatively resemble the results presented here. However, the size-dependencies of AGR and FGR observed in our simulations cannot be generalized for arbitrary substances and environments, as they are affected by the evaporation profile of the substance, and the time evolution of the cluster distribution, determined by the time evolution of the sources and sinks. For example, in the atmosphere the monomer source rate is usually time-dependent, which may affect the size-dependence of the growth rates (see Olenius et al. (2014) where this case is discussed).

To clarify this, we added the following sentence in the Sect. 3.1 (page 9, line 24):

It needs to be noted that the size-dependencies of AGR and FGR observed here cannot be generalized for arbitrary substances and environments, because they are affected by the vapor properties as well as the ambient conditions.

\* Finally, I think that the discussion in the conclusions could be improved. In the last paragraph, (p. 14, L 23) the authors state that "conventional methods used to determine particle formation rates from growth rates may give estimates far from their true values". Based on earlier discussion, it seems that the authors bring forward the idea that not a single 'true' value exists (for example, the growth rate can mean the AGR, FGR or CGR). In the study of particle formation, finding the values for the formation rate J and the growth rate GR has for some time been a key objective. It would be useful if the authors could discuss here which of the given growth rates is the true value meant.

We agree that the sentence quoted by the referee was not entirely clear, and it has now been slightly changed. The aim of the statement is not to suggest that any of these growth rates would be the true growth rate of particles. Indeed, in the cases discussed in this study, where clusters grow not only by monomer collisions but also by cluster-cluster collisions, it is not possible to give one, unambiguous value for the growth rate of clusters of a specific size. With the "true value", we refer to the true value of the particle formation rate J, i.e. the flux of particles past a certain size, that is always unambiguously defined. We want to point out that determining a formation rate from a measured growth rate (here AGR) utilizing Eq. (5), as is often done, may lead to values far from the real particle flux. This issue is discussed, for example, in the end of the Sect. 3.1.

**SPECIFIC COMMENTS**

P7 L33: Regarding the monomer concentration ranges, I understand the limitation of the lower level. However, the upper level for the monomer concentration is less justified. This is a case of selecting a too long time step; reducing the time step in the differential equation solving would present usable results. In any case, the limits for the usability of the methods should be given.

Actually, the upper limit for the monomer concentration is not determined by the time step of the simulations, but the width of the size bins, for which the growth rates are determined. As clarified in the end of the Sect. 2.2 (page 8, line 2), if the monomer concentration was set to a too high value, concentrations of large clusters become so high that a significant fraction of the flux from a certain size bin may end up not only in the next size bin but also in the size bins larger than that. This would complicate the growth dynamics, and the method used to calculate the flux-equivalent growth rate (see Sect. 2.1.1) would not be exactly valid (see also the reply to a comment on Eq. (1) by Referee #1). Also, determining AGR from the appearance times of two consecutive bins (Eq. (4)) would not be justified if significant fractions of the flux from bin *i* would end up also in bins > i+1. As we are interested in situations where the clusters grow mainly by collisions of vapor monomer and small clusters, we have not addressed cases where the self-coagulation of the population becomes significant. We have now added a reference to Table 1 when discussing the limitations of the methods (page 8). In general, the limits for the validity of these methods are determined by the combination of all the parameters in the simulations (e.g. monomer source rate, the magnitude of the external sink, the properties of model substance) and the size bin width used in the analysis, and are thus systemspecific.

P19, Table 1: For reproducibility, it is not enough to give the final monomer concentration, but also the monomer source rate is important information. I think that the monomer source rates used to obtain the final concentrations in Table 1 should be given also in Table 1.

As the referee suggests, we now added the monomer source rates used in different simulations into Table 1.

**Growth of atmospheric clusters involving cluster-cluster collisions: comparison of different growth rate methods**

J. Kontkanen1, T. Olenius2, K. Lehtipalo1,3, H. Vehkamäki1, M. Kulmala1 and K. E. J. Lehtinen4

[revised manuscript text omitted]

(5)

In the analysis of experimental particle size distribution data, Eq. (5) is generally used to calculate the particle flux towards larger sizes from measured or calculated growth rates (Kulmala et al., 2012). Thus, comparing the flux  $J_{app}$  to the real flux J, which is directly obtained from the simulations, provides information on how the use of Eq. (5) affects the conclusions of the data analysis.

In addition to the change in the mass diameter of the clusters, we determine FGR and AGR also as the change in the number of molecules,  $N_{\text{mols}}$ . These are obtained by replacing the width of the size bin  $\Delta D_{\text{p},i}$  in diameter space in Eqs (3) and (4) with the width of the bin  $\Delta N_{\text{mols}}$  in number of molecules space.

**2.1.3 Growth rate by assuming irreversible vapor condensation (CGR)**

35 The kinetic hard-spheres collision rate between a vapor molecule with diameter  $D_{mon}$  and a cluster with diameter  $D_p$  is given by

$$K_{\rm kin} = \frac{\pi}{4} (D_{\rm p} + D_{\rm mon})^2 (\bar{c}_{\rm p}^{\ 2} + \bar{c}_{\rm mon}^{\ 2})^{1/2} (C_{\rm mon} - C_{\rm e}), \tag{6}$$

where  $\bar{c}$  is the thermal speed,  $C_{\text{mon}}$  is the vapor concentration and  $C_{\text{e}}$  is the vapor concentration corresponding to the equilibrium vapor pressure over the cluster. When analyzing experimental data, it is usually assumed that saturation vapor

pressure  $p_{sat} = 0$ , which means that the vapor is assumed to condense irreversibly. In this case, the growth rate of a cluster as a change of mass diameter is obtained from (Nieminen et al., 2010):

$$CGR = \frac{\gamma}{2\rho} \left(1 + \frac{D_{\text{mon}}}{D_{\text{p}}}\right)^2 \left(\frac{3k_{\text{B}}T}{\pi}\right)^{1/2} \left(\frac{1}{m_{\text{p}}} + \frac{1}{m_{mon}}\right)^{1/2} m_{\text{mon}} C_{\text{mon}}.$$
(7)

Here  $\rho$  is the condensed phase density, and  $m_p$  and  $m_{mon}$  are the masses of the cluster and the vapor molecule.  $\gamma$  is a correction

5 factor that needs to be added if CGR is calculated in continuum regime (see Nieminen et al., 2010). When applying Eq. (7) in this study for calculating CGR for different size bins, we use the diameter and mass of the cluster at the upper limit of each size bin for  $D_p$  and  $m_p$ .

In addition to the fact that CGR calculated from Eq. (7) takes into account only monomer collisions and no evaporation, the essential difference between CGR and FGR is the perspective from which the growth is studied. CGR corresponds to the

10 traditional Lagrangian approach, where the growth of an individual cluster between different size bins is followed. FGR, on the other hand, corresponds to the Eulerian approach, where the net flux between adjacent size bins is studied. See Olenius et al. (2014) for further discussion about the differences between these approaches.

**2.2 Simulations**

We simulated the time evolution of cluster concentrations in a one-component system using the Atmospheric Cluster Dynamics

- 15 Code (ACDC; McGrath et al., 2012; Olenius et al., 2014). The model included the production of monomers, all the possible collision and evaporation processes between different clusters, and the losses of clusters due to an external sink. The model substance was assumed to consist of spherical molecules and clusters with the properties of sulfuric acid: a molecular mass  $(m_{mon})$  of 98.08 amu, a liquid density ( $\rho$ ) of 1830 kg m-3 and a surface tension ( $\sigma$ ) of 0.05 N m-1. However, the saturation vapor pressure of the model substance was lowered from that of sulfuric acid to decrease the evaporation rate of clusters. This
- 20 qualitatively mimics the stabilization of sulfuric acid clusters by base molecules, such as ammonia or amines (e.g. Kurtén et al., 2008). The simulations included clusters with 1 to 70 molecules; the clusters growing larger than that were assumed to be stable and removed from the simulation. The Gibbs free energy of formation of the clusters was calculated from the classical one-component liquid droplet model to obtain a qualitatively realistic evaporation profile. In the majority of the simulations the Gibbs free energy profile had a single maximum and no minima as a function of cluster size. This corresponds to a
- 25 monotonically increasing stability with the increasing cluster size in the studied size range. In addition, a set of simulations was performed using a free energy profile with lowered formation free energies for the smallest clusters (see Fig. A1 in Appendix). This corresponds to a system with elevated concentrations of small stable clusters, similarly as in the simulation study by Vehkamäki et al. (2012) and possibly also in the atmosphere (Kulmala et al., 2013). Collision rates between clusters were obtained from Eq. (6) with  $C_e = 0$ , and cluster evaporation rates were calculated from the Gibbs free energies of formation
- 30 of the clusters (e.g. Ortega et al., 2012). The external losses were assumed to depend on the cluster size according to (Lehtinen et al., 2007):

$$S(D_{\rm p}) = S(D_{\rm mon}) \times \left(\frac{D_{\rm p}}{D_{\rm mon}}\right)^b \tag{8}$$

[revised manuscript text omitted]
- 20 U., Riipinen, I., Curtius, J., Worsnop, D. R., and Kulmala, M.: The effect of acid-base clustering and ions on the growth of atmospheric nano-particles, Nature Communications, 2016, in review.

McGrath, M. J., Olenius, T., Ortega, I. K., Loukonen, V., Paasonen, P., Kurtén, T., Kulmala, M., and Vehkamäki, H.: Atmospheric Cluster Dynamics Code: a flexible method for solution of the birth-death equations, Atmos. Chem. Phys., 12, 2345-2355, doi:10.5194/acp-12-2345-2012, 2012.

25 Manninen, H. E., Nieminen, T., Riipinen, I., Yli-Juuti, T., Gagné, S., Asmi, E., Aalto, P. P., Petäjä, T., Kerminen, V.-M., and Kulmala, M.: Charged and total particle formation and growth rates during EUCAARI 2007 campaign in Hyytiälä, Atmos. Chem. Phys., 9, 4077-4089, doi:10.5194/acp-9-4077-2009, 2009.

Merikanto, J., Spracklen, D. V., Mann, G. W., Pickering, S. J., and Carslaw, K. S.: Impact of nucleation on global CCN, Atmos. Chem. Phys., 9, 8601–8616, doi:10.5194/acp-9-8601-2009, 2009.

30 Nieminen, T., Lehtinen, K. E. J., and Kulmala, M.: Sub-10 nm particle growth by vapor condensation – effects of vapor molecule size and particle thermal speed, Atmos. Chem. Phys., 10, 9773-9779, doi:10.5194/acp-10-9773-2010, 2010.

Olenius, T., Riipinen, I., Lehtipalo, K., and Vehkamäki, H.: Growth rates of atmospheric molecular clusters based on appearance times and collision–evaporation fluxes: Growth by monomers, J. Aerosol Sci., 78, 55–70, 2014.

Ortega, I. K., Kupiainen, O., Kurtén, T., Olenius, T., Wilkman, O., McGrath, M. J., Loukonen, V., and Vehkamäki, H.: From
quantum chemical formation free energies to evaporation rates, Atmos. Chem. Phys., 12, 225-235, doi:10.5194/acp-12-225-2012, 2012.

Seinfeld, J. H. and Pandis, S. N.: Atmospheric chemistry and physics: from air pollution to climate change, 2nd Edn., John Wiley & Sons, Inc., Hoboken, NJ, 2006.

Sihto, S.-L., Kulmala, M., Kerminen, V.-M., Dal Maso, M., Petäjä, T., Riipinen, I., Korhonen, H., Arnold, F., Janson, R., Boy, M., Laaksonen, A., and Lehtinen, K. E. J.: Atmospheric sulphuric acid and aerosol formation: implications from

5 atmospheric measurements for nucleation and early growth mechanisms, Atmos. Chem. Phys., 6, 4079-4091, doi:10.5194/acp-6-4079-2006, 2006.

Spracklen, D. V., Carslaw, K. S., Kulmala, M., Kerminen, V.-M., Sihto, S.-L., Riipinen, I., Merikanto, J., Mann, G. W., Chipperfield, M. P., and Wiedensohler, A.: Contribution of particle formation to global cloud condensation nuclei concentrations, Geophys. Res. Lett., 35, L06808, doi:10.1029/2007GL033038, 2008.

10 Wang, J., McGraw, R. L., and Kuang, C.: Growth of atmospheric nano-particles by heterogeneous nucleation of organic vapor, Atmos. Chem. Phys., 13, 6523-6531, doi:10.5194/acp-13-6523-2013, 2013.

Vanhanen, J., Mikkilä, J., Sipilä, M., Manninen, H. E., Lehtipalo, K., Siivola, E., Petäjä, T., and M., K. Particle Size Magnifier for Nano-CN Detection. Aerosol Sci. Technol. 45:533–542, 2011.

Weber, R. J., Marti, J. J., McMurry, P. H., Eisele, F. L., Tanner, D. J., and Jefferson, A.: Measurements of new particle formation and ultrafine particle growth rates at a clean continental site, J. Geophys. Res., 102, 4375–4385, 1997.

Vehkamäki, H., McGrath, M. J., Kurtén, T., Julin, J., Lehtinen, K. E. J., and Kulmala, M.: Rethinking the application of the first nucleation theorem to particle formation, J. Chem. Phys., 136, 094107, doi:10.1063/1.3689227, 2012.

Vuollekoski, H., Sihto, S.-L., Kerminen, V.-M., Kulmala, M., and Lehtinen, K. E. J.: A numerical comparison of different methods for determining the particle formation rate, Atmos. Chem. Phys., 12, 2289-2295, doi:10.5194/acp-12-2289-2012, 2012

Yli-Juuti, T., Nieminen, T., Hirsikko, A., Aalto, P. P., Asmi, E., Hõrrak, U., Manninen, H. E., Patokoski, J., Dal Maso, M., Petäjä, T., Rinne, J., Kulmala, M., and Riipinen, I.: Growth rates of nucleation mode particles in Hyytiälä during 2003–2009: variation with particle size, season, data analysis method and ambient conditions, Atmos. Chem. Phys., 11, 12865-12886, doi:10.5194/acp-11-12865-2011, 2011.

25

20

15

| Simulation set 1: Varying the monomer source rate                                            |                                    |                                  |                                    |
|----------------------------------------------------------------------------------------------|------------------------------------|----------------------------------|------------------------------------|
| Monomer source rate (cm -3 s -1 )                                      | 1.0×10 3                | 5.5×10 3              | 1.8×10 4                |
| Steady-state monomer concentration (cm -3 )                                       | $1 \times 10^{6}$                  | $5 \times 10^{6}$                | $1 \times 10^{7}$                  |
| Loss coefficient (s -1 )                                                          | 10 -3 s -1   | 10 -3 s -1 | 10 -3 s -1   |
| Saturation vapor pressure (Pa)                                                               | 2×10 -10                | 2×10 -10              | 2×10 -10                |
| Size bin width (molecules)                                                                   | 10                                 | 10                               | 10                                 |
| Simulation set 2: Varying the saturation vapor pressure                                      |                                    |                                  |                                    |
| Monomer source rate (cm -3 s -1 )                                      | <mark>5.8×103</mark>    | $5.5 \times 10^{3}$              | 5.1×10 3                |
| Steady-state monomer concentration (cm -3 )                                       | $5 \times 10^{6}$                  | $5 \times 10^{6}$                | $5 \times 10^{6}$                  |
| Loss coefficient (s -1 )                                                          | 10 -3 s -1   | 10 -3 s -1 | 10 -3 s -1   |
| Saturation vapor pressure (Pa)                                                               | 1.5×10 -10              | 2×10 -10              | 1×10 -9                 |
| Size bin width (molecules)                                                                   | 10                                 | 10                               | 10                                 |
| Simulation set 3: Varying the loss coefficient                                               |                                    |                                  |                                    |
| Monomer source rate (cm -3 s -1 )                                      | <mark>5.5×103</mark>    | 5.5×10 3              | 5.5×10 3                |
| Steady-state monomer concentration (cm -3 )                                       | $6.4 \times 10^{6}$                | $5 \times 10^{6}$                | $2.6 \times 10^{6}$                |
| Loss coefficient (s -1 )                                                          | 7×10 -4 s -1 | 10 -3 s -1 | 2×10 -3 s -1 |
| Saturation vapor pressure (Pa)                                                               | 2×10 -10                | 2×10 -10              | 2×10 -10                |
| Size bin width (molecules)                                                                   | 10                                 | 10                               | 10                                 |
| Simulation set 4: Varying the size bin width                                                 |                                    |                                  |                                    |
| Monomer source rate (cm -3 s -1 )                                      | 5.5×10 3                | 5.5×10 3              | 5.5×10 3                |
| Steady-state monomer concentration (cm -3 )                                       | $5 \times 10^{6}$                  | $5 \times 10^{6}$                | $5 \times 10^{6}$                  |
| Loss coefficient (s -1 )                                                          | 10 -3 s -1   | 10 -3 s -1 | 10 -3 s -1   |
| Saturation vapor pressure (Pa)                                                               | 2×10 -10                | 2×10 -10              | 2×10 -10                |
| Size bin width (molecules)                                                                   | 5                                  | 10                               | 14                                 |
| Simulation set 5: Varying the monomer source rate with a different Gibbs free energy profile |                                    |                                  |                                    |
| Monomer source rate (cm -3 s -1 )                                      | $1.1 \times 10^{3}$                | <mark>9.6×103</mark>  | 5.5×10 4                |
| Steady-state monomer concentration (cm -3 )                                       | $1 \times 10^{6}$                  | $5 \times 10^{6}$                | $1 \times 10^{7}$                  |
| Loss coefficient (s -1 )                                                          | 10 -3 s -1   | 10 -3 s -1 | 10 -3 s -1   |
| Saturation vapor pressure (Pa)                                                               | 2×10 -10                | 2×10 -10              | 2×10 -10                |
| Size bin width (molecules)                                                                   | 10                                 | 10                               | 10                                 |

ie 1. Summary of the performed simulations.